# Predicting monoclonal antibody binding sequences from a sparse sampling of all possible sequences
Pritha Bisarad [1,2,3,4,10], Laimonas Kelbauskas[5,6,10], Akanksha Singh [1,2,7,10], Alexander T. Taguchi [8], Olgica Trenchevska[9] & Neal W. Woodbury [1,2,5] ✉

Previous work has shown that binding of target proteins to a sparse, unbiased sample of all possible peptide sequences is sufficient to train a machine learning model that can then predict, with statistically high accuracy, target binding to any possible peptide sequence of similar length. Here, highly sequence-specific molecular recognition is explored by measuring binding of 8 monoclonal antibodies (mAbs) with specific linear cognate epitopes to an array containing 121,715 near-random sequences about 10 residues in length. Network models trained on resulting sequence-binding values are used to predict the binding of each mAb to its cognate sequence and to an in silico generated one million random sequences. The model always ranks the binding of the cognate sequence in the top 100 sequences, and for 6 of the 8 mAbs, the cognate sequence ranks in the top ten. Practically, this approach has potential utility in selecting highly specific mAbs for therapeutics or diagnostics. More fundamentally, this demonstrates that very sparse random sampling of a large amino acid sequence spaces is sufficient to generate comprehensive models predictive of highly specific molecular recognition.

There are many examples of using machine learning approaches to analyze or engineer protein structure, binding or function starting with amino acid sequence (reviewed previously[1–4]). Past work from this lab has focused on a simple sequence-binding system based on large libraries (>100,000 unique sequences) of essentially random sequence peptides with lengths of about 10 amino acids. These peptides are synthesized in high density arrays on silica surfaces using photolithographic approaches common in the electronics industry[5]. The arrays are made commercially by the hundreds with extremely high reproducibility, and thus represent a convenient and inexpensive tool for developing sequence-binding relationships for large numbers of samples. Initially, the binding of a set of nine different proteins was analyzed[6]. The fluorescently labeled proteins were incubated with the arrays and the binding to each sequence was recorded. A fraction of the peptide array data was used to train a fully connected neural network resulting in a model which was then used to predict the binding to the remaining sequences on the array. The sequences not used for training represented an independent and nearly random test sample set. Because the sequences are essentially random, this implied that the approach was able to create statistically accurate models for the binding of each of the nine diverse proteins to any of the roughly trillion possible amino acid sequences in this length range (correlations of measured vs. predicted binding typically >0.95). A similar approach was then applied to total IgG in serum for a group of >500 individuals from 6 different cohorts (4 cohorts with viral infections, one cohort with Chagas disease and one with no known infections)[7]. Again, it was possible to predict IgG binding for each individual to any sequence in this length range with high statistical accuracy. Further, the disease specific information was captured by the model such that the ability to distinguish between cohorts using either supervised classification our unsupervised clustering was more accurate when using the model output than using the raw data, particularly in the presence of sequence independent noise in the measured data. Recently, the approach has been applied to understanding the humoral immune response to Lyme Disease using similar approaches[8].

Peptide binding to protein surfaces or to total IgG associated with polyclonal immune responses both involve many possible interactions

[1]School of Molecular Sciences, Arizona State University, Tempe, AZ, USA. [2]Center for Innovations in Medicine, Biodesign Institute, Arizona State University, Tempe, AZ, USA. [3]Pediatric Movement Disorders Program, Division of Pediatric Neurology, Barrow Neurological Institute, Phoenix Children's Hospital, Phoenix, AZ, USA. [4]Department of Child Health, University of Arizona College of Medicine-Phoenix, Phoenix, AZ, USA. [5]Center for Molecular Design and Biomimetics, Biodesign Institute, Arizona State University, Tempe, AZ, USA. [6]Biomorph Technologies, Chandler, AZ, USA. [7]Prellis Biologics Inc., Berkeley, CA, USA. [8]iBio Inc., San Diego, CA, USA. [9]Cowper Sciences Inc., Chandler, AZ, USA. [10]These authors contributed equally: Pritha Bisarad, Laimonas Kelbauskas, Akanksha Singh. ✉e-mail: nwoodbury@asu.edu

between the target or sample and each sequence. Thus, what is modeled is an average response and this is generally more continuous in terms of sequence variation vs. binding affinity than one might expect for a single, highly specific interaction between a target and a particular amino acid sequence. It is likely that developing models that recognize interactions with very specific sequences based on binding measurements to a sparce, random set of peptide sequences would be much more challenging. Here, the ability of such peptide array-based models to predict the binding of a monoclonal antibody (mAb) to a particular linear cognate sequence that it is known to bind with high specificity and high affinity is examined. This is performed by training a neural network model on the binding levels of each of eight such mAbs to 121,715 peptide sequences chosen nearly randomly and synthesized in an array format. The resulting models are then used to compare the predicted binding level of the cognate sequence for each of the eight mAbs to the predicted binding levels of a set of one million randomly selected sequences. If the model is accurate, one would both expect that it would predict a binding affinity for the cognate sequence that ranks near the top of the list of binding levels for the million random sequences and additionally that it would be able to accurately predict which of the amino acids in the cognate sequence were essential to binding.

The primary goal of this work is to explore whether measurements of the binding of a mAb to a sparse, random sample of combinatorial sequence space can be used to train a machine learning model to recognize the very specific sequence-binding interactions of that mAb. However, descriptions of mAb binding are of potential practical interest as well due to their broad applications in research, diagnostics and therapeutics. Currently there are more than 150 approved mAbs in use therapeutically or under review[9] and many more involved in diagnostic tests and as research reagents. The key qualities of mAbs are their affinity and specificity, with mAbs often having dissociation constants of tens to hundreds of picomolar with relatively little off-target binding[10]. Binding typically involves interactions between the mAb and amino acids in a specific spatial arrangement[11]. This can be the particular order of amino acids in a linear sequence (continuous epitopes) or their arrangement at the surface of a protein, potentially from multiple noncontiguous regions (discontinuous or structural epitopes). The multiple interactions at defined positions is what affords the mAb both high affinity and high specificity.

Many experimental and computational tools exist to explore mAb binding interactions. Experimentally, X-ray crystallography can provide the highest structural resolution of antibody-antigen binding, but is not suited

for exploring weaker binding interactions[12] or very large numbers of mAb-antigen pairs. Cross-linking mass spectrometry can provide insight into off-target binding interactions, but lacks the resolution and sensitivity to quantitate relative binding affinities[13,14]. Peptide microarrays and bead libraries consisting of tiled protein or whole proteome sequences can be used to determine binding sites of antibodies, but are generally specific to the sequences tiled and, for whole proteomes, may require either very large, or rather specialized, libraries[15,16], though generalized approaches have also been developed[17]. Computationally, the complexity of these interactions has made predictive modeling of mAb binding and epitope prediction a challenging problem[18]. Molecular docking is a useful computational tool to estimate antibody-antigen binding sites but can be computationally expensive when screening a mAb against a large library of potential targets and usually does not take advantage of empirical binding data[19–21]. Epitope binding sites can be predicted by modeling epitope databases or antibody-antigen structures, but these predictions are generally not specific to a given mAb[19,22,23]. An important application of these approaches is the application to predicting off-target binding[24,25], a significant issue particularly with regard to therapeutic mAbs[14,26]. While the approaches described above provide important insight into the specific binding interactions mAbs engage in, most face practical limitations in scaling to large numbers of mAbs and large numbers of potential targets or off-targets.

In this work, a set of eight mAbs with well-characterized linear epitopes are bound to a planar array of near-random peptide sequences 5–11 residues in length. The binding values are then used to train a neural network generating a comprehensive and quantitative model of mAb binding which can be used to predict the binding of the mAb to any amino acid sequence of similar length. The ability of that model to recognize the known, specific cognate binding sequence of the mAb is then explored.

## Results and discussion
### Monoclonal antibodies used
The eight monoclonal antibodies (mAb) used in this study are listed in Table 1. All of these mAbs have known, linear epitopes of contiguous amino acids (for Ab8[27], 4C1[28], Lnkb2[29] the antigen analysis is published and for others epitope and antigen information is provided by the supplier listed in Table 1). All were raised to human targets except for Ab1, and all are Murine antibodies. In each case, 4 concentrations of the mAb (0.125 nM, 0.5 nM, 2.0 nM, 8.0 nM) were used in binding assays and the binding measurements at each concentration were replicated 4 times.

**Table 1 | Monoclonal antibodies used**

| Name | Target[a] | Epitope use for evaluation[b] | Array sequences[c] | Source |
|------|-----------|-------------------------------|--------------------|--------|
| Ab1 | aa 211-220 Murine p53-beta galactosidase fusion protein expressed in E. coli., P04637 | RHSVVVP | RHSVV, RHSVVV | Millipore/Sigma Cat# CBL404 |
| Ab8 | Bacterially expressed full-length human p53, P04637 | SDLWKLL | SDLWKLL, SDLWKL | ThermoFisher Cat# MA1-19055 |
| 4C1 | External domain of human TSH receptor., P16473 | QAFDSHY | LQAFDS, QAFDSH, FDSHYD | GeneTex Cat# GTX47974 |
| DM1A | aa 426-450 - human brain α-Tubulin., Q71U36 | LEKDYEE | AALEKDY, ALEKDYE, LEKDYEE | Millipore/Sigma Cat# 05-829 |
| Lnkb2 | Human IL-2, P60568 | PLEEVLN | PLEEVLN | Absolute Antibody Cat# Ab00232-1.1 |
| TF3B5 | Aa 1242-1255 C-terminus human ErbB-2 (HER2), P04626 | PEYLGLD | None | ThermoFisher Cat# Ma5-13675 |
| C3 | aa 251-450 N-terminal extracellular domain human ErbB-2 (HER2), P04626 | SLPNPEG | None | Santa Cruz Bio Cat# SC-377344 |
| 9E10.3 | aa 408-439 C-terminus of human c-myc, P01106 | KLISEED | None | ThermoFisher Cat# MA5-12080 |

[a]The targets/antigens the mAbs were raised to. Each target description is followed by a uniport reference number
[b]These epitopes were used as the cognate epitope sequence in determining the rank of binding out of 1 million random sequences, see Table 2. Note that because there is no isoleucine on the array, I was substituted for V in mAb 9E10.3 when predicting ranks.
[c]The sequences given here were those purposely synthesized on the peptide array in order to evaluate binding to the cognate epitope. TF3B5, C3 and 9E10.3 epitopes were not on the array. Cognate sequences were excluded from the dataset used to train the neural network models.

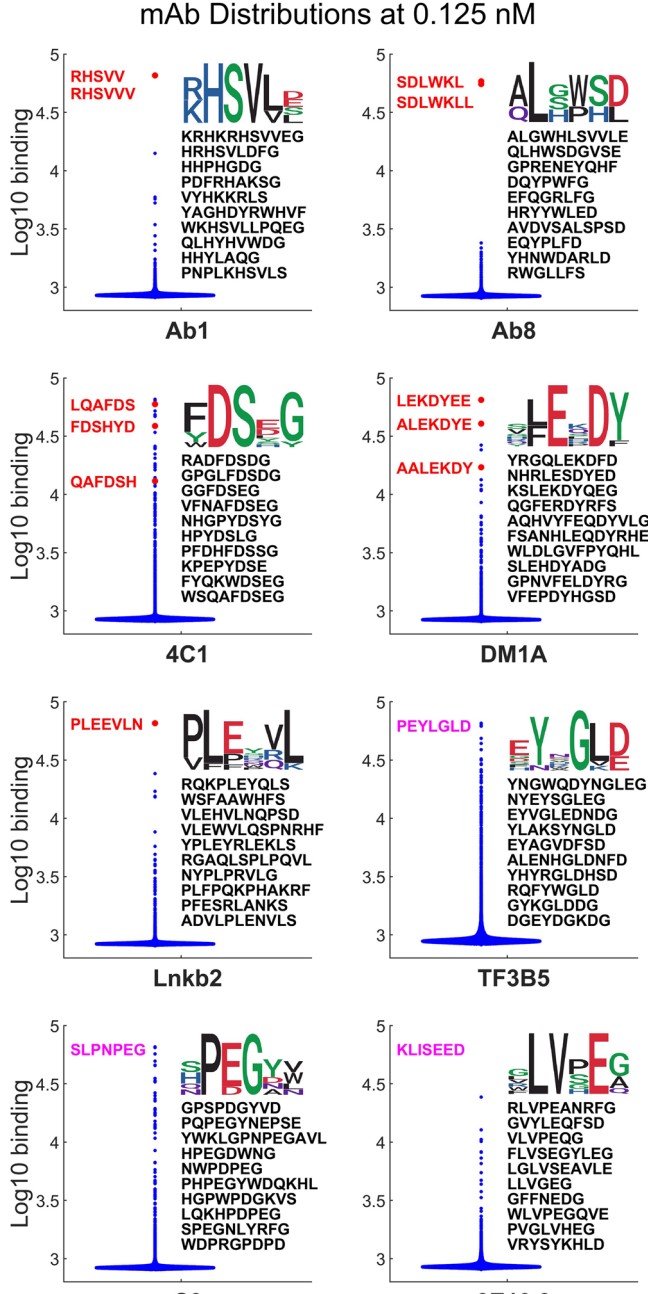

**mAb Distributions at 0.125 nM**

**Fig. 1 | Violin plots of log 10 binding.** Blue dots: array peptides (121,715 unique sequences), Red dots: cognate sequences. Cognate sequences represented on the array are shown in red, top ten array sequences are in black. Cognate sequences in magenta are not synthesized on the array. The SeqLogo derived from the top ten sequences is shown above them. See Table S1 for numbers of peptides >2 standard deviations above the median.

## Peptide array assay

Monoclonal antibody preparations were incubated with peptide arrays consisting of 126,051 unique peptide sequences (of which 121,715 were in the size range used) that were selected to sample all possible amino acid sequence combinations of approximately 10 amino acids as evenly as possible (a complete description of the size distribution and amino acid distribution as a function of residue position has been published previously[6]). These peptides were built using 16 of the 20 natural amino acids, A D E F G H K L N P Q R S V W and Y. Note that with the exception of 9E10.3, all of the mAbs had cognate epitope binding sequences that only included these amino acids (9E10.3 contains an isoleucine and for the analyses below, this

was assumed to be similar to valine). Control sequences distributed across the arrays were used to gauge the uniformity of binding and multiple copies of cognate epitopes and some variants were synthesized on the peptide array for 5 of the 8 mAbs (Table 1). The mAb cognate epitope sequences were used to gauge the actual binding level of the mAb to its cognate epitope in the context of an isolated linear sequence attached to substrate. The binding of each mAb to the peptide arrays was performed as described in Methods and binding of a fluorescently labeled secondary antibody was used to determine how much mAb was bound to each sequence by detecting with an array scanner (Methods).

The four replicates of each mAb at each concentration were averaged as the mean of the $\log_{10}$ values of the observed binding. Based on the comparisons of Pearson correlation coefficients between replicates (Supplementary Fig. S1), no replicate arrays were excluded from the averaging. Based on the standard deviations between replicates for each binding value in each array, no individual binding values were excluded from the averaging; all collected data was used without removing outliers. Except where noted below, the average $\log_{10}$ values from the replicates were used for analysis and all modeling of the sequence-binding relationships was performed after removing the mAb cognate sequences from the array dataset to avoid training on those epitope sequences.

### Distributions of binding to array sequences

Figure 1 and Supplementary Fig. S2 show the distribution of binding values for each mAb to the array sequences in different ways after averaging replicates. Supplementary Fig. S2 provides a plot of the $\log_{10}$ number of peptides at each binding value for each mAb and concentration. Figure 1 focuses on the lowest concentration, comparing violin plots of the distribution. Note that the y-axis is $\log_{10}$ binding. The cognate sequence binding of each mAb (sometimes several versions) are represented as red dots for the first five mAbs and the cognate sequence itself is given in red letters for those mAbs and in magenta for the final three mAbs. In all but one case (4C1) at least one version of the mAb cognate sequence saturates the detector (65,536 counts or 4.82 on a $\log_{10}$ scale). The top ten binding array sequences and the SeqLogo consensus sequence[30] determined from them is shown.

Several things are evident in Fig. 1. First, as a result of the specificity of mAbs the vast majority of sequences bind at values very close to the background (about 850 counts or 2.9 on the log scale). For most mAbs there are only a handful of points higher than 3.5 and for Ab8 there are none. Supplementary Table S1 shows how many peptides have values that are above background based on the variation seen in the replicates. Second, in most cases, the top ten sequences define some identifiable motif related to the known cognate sequence. While this is less evident for Ab8, one sees that the amino acid composition of the top sequences is similar to that of the cognate epitope. Thus, there is antibody-specific information in the binding to even a sample of near-random peptide sequences. Finally, in most cases there is also one or more apparently unrelated sequences in the top ten that may represent a mimotope, i.e. a peptide that, despite apparent dissimilarity in the letter code, can have similar physicochemical properties to the cognate sequence. The binding distributions of the mAbs exemplify the challenge of using binding data from peptides that randomly sample all of the 16 amino acid sequence space ($16^{10} = \sim 10^{12}$ sequences) without regard to any specific biological relevance to create comprehensive sequence-binding models. While this approach ensures generality as it is not biased towards any particular outcome, the specificity of mAbs means that the bulk of the data from the array is from sequences that bind very weakly.

### Modeling the sequence-binding relationship using machine learning methods

To build a general relationship between sequence and mAb binding from the array data, a fully connected neural network was employed with 2 hidden layers and 250 nodes per layer. This was similar in structure to networks used previously for describing serum IgG binding[7] or binding to isolated proteins[6]. Two additions were made to the sequence representation

**Table 2 | Ranks out of 1,000,000 random sequences based on concentration**

| mAb | Epitope | Baseline[a] | +Weighting[b] | +Shifting[c] | +Comp[d] | % Rank[e] |
|---|---|---|---|---|---|---|
| Ab1 | RHSVVVP | 280,000 ± 30,000 | 580 ± 100 | 2.6 ± 0.7 | 3.0 ± 0.7 | 0.0003% |
| Ab8 | SDLWKLL | 680,000 ± 90,000 | 720,000 ± 50,000 | 16 ± 2 | 7.9 ± 1.2 | 0.0008% |
| 4C1 | QAFDSHY | 700 ± 200 | 120 ± 20 | 12 ± 2 | 15 ± 1.5 | 0.0015% |
| DM1A | LEKDYEE | 1.6 ± 0.6 | 29 ± 4 | 1.8 ± 0.4 | 2.2 ± 0.5 | 0.0002% |
| Lnkb2 | PLEEVLN | 22,000 ± 11,000 | 61 ± 25 | 1.4 ± 0.2 | 1.4 ± 0.2 | 0.0001% |
| TF3B5 | PEYLGLD | 11 ± 2.6 | 10 ± 0.7 | 1.0 ± 0.0 | 1.1 ± 0.1 | 0.0001% |
| C3 | SLPNPEG | 48 ± 9 | 10 ± 2 | 9.0 ± 1.9 | 9.5 ± 1.4 | 0.0010% |
| 9E10.3 | KLVSEED | 340 ± 90 | 160 ± 30 | 2400 ± 500 | 79 ± 3.9 | 0.0079% |

[a]This is the baseline case without weighting high binding values, shifting the sequence register or normalizing for composition-dependent binding. The standard error is shown.
[b]Baseline case plus additional weighting of those sequences that bound the mAb with high affinity
[c]Weighting plus adding 5 copies of each sequence in shifted register, as described in the text
[d]Weghting, shifting and normalization by subtracting the log of the composition dependent binding from the log of the total binding (full model)
[e]Based on rank after weighting, shifting and composition adjustments.

beyond what has been done previously. Peptide sequences with fewer than the maximum number of residues were padded with a "blank" residue (marked as "X"), so that they were all the same length. Also, the N- and C-termini were marked with start and end tokens "(" and ")", respectively. The sequences, including the three additional characters, were then input into the neural network as one-hot encoded vectors, similar to previous work[6,7]. Note that only the peptides between 5 and 11 residues in length were used in the training (there were 121,715 unique peptides in this length range) because the focus was on identifying continuous epitopes which are generally in that length range and because there is a tendency for the neural network to overestimate the binding of longer peptides, simply due to the larger number of elements to assign value to, potentially biasing the model.

One other adjustment was made to the dataset prior to running the neural network. At higher concentrations, for some of the mAbs, there were a number of sequences whose binding saturated the detector (Supplementary Fig. S2), potentially causing the neural network to underestimate high binding values. This was corrected by substituting saturating values at one concentration with the value from the next lower concentration multiplied by the relative increase in concentration. This assumption of linearity with concentration is justified by the fact that binding values of peptides on the array in the linear range, do, in fact, increase linearly with concentration (Supplementary Fig. S3). Each mAb was modeled separately, but all four concentrations for the mAb were used as targets simultaneously. Thus, the final weight matrix of the neural network had four columns, each associated with a different concentration.

**Evaluating the neural network prediction performance**
For performing the binding predictions below, the neural network was trained 12 separate times using the binding of all measured array sequences to the mAbs in the training set (except for the cognate sequences which were removed during training). (For reference, Supplementary Fig. S4 shows the results of a 10-fold cross validation of the sequence-binding relationship for each mAb to illustrate the accuracy of predictions within the measured dataset.) After training, for each mAb, the resulting models were used to predict the binding of the mAb at each concentration to one million in silico randomly generated sequences of the same length as the cognate sequence and the values of the 12 models were averaged (cognate sequences used given in Table 1). The average binding value predicted from the 12 models for the cognate sequence was then compared to those of the million random sequences and the rank position of the cognate sequence was determined (Table 2, details provided in the supplementary material). The ranking process described above was repeated 10 times with a different million random sequences each time and the error of the mean of each cognate sequence rank was determined. As a result, both variation due to different random sequence sets and variation as a result of different random starting points for the neural network training are taken into account in the data and errors provided.

**Baseline prediction**
Table 2, column 2 gives the rank of the predicted cognate epitope sequence based on the binding as described above using the input sequences with no further modification. Five of the eight mAb cognate sequences (Ab1, 4C1, DM1A, TF3B5, C3) are in the top 0.1% (top 1000) of the random sequences, and two of them, DM1A and TF3B5, are in the top ~0.001%. The prediction for both Ab1 and Ab8 is essentially random. While being in the top 0.1% implies this simple modeling approach predicts significant specificity of mAb binding, the specificity is not at the level suggested by the binding of the cognate sequence in Fig. 1, particularly for Ab1 and Ab8.

**Weighting the highest value binding sequences**
It is clear from Fig. 1 and Supplementary Table S1 that the number of sequences that bind substantially to the monoclonal antibodies is generally small. Yet it is the higher binding sequences that contain some of the most important binding information. It thus makes sense to balance the dataset by weighting high binding values more than the very large number of low binding values in the modeling. To accomplish this, binding distributions of the 8 nM mAbs were used to develop a weighting rule and weighting was implemented by increasing the number of copies of the high binding sequences accordingly (see Methods). When the same neural network model was applied to the weighted dataset, predictions for several of the mAbs improved markedly. In particular Ab1, 4C1, and Lnkb2 improved their ranks in the million random sequences by roughly one to three orders of magnitude. Ab8 did not improve with weighting and the DM1A cognate sequence actually became significantly less highly ranked.

**Training on multiple copies of each sequence with shifted registers**
In the modeling described above, the residue positions of each sequence are in a specific register within the one-hot vector describing that sequence. However, it would be desirable for the neural network to learn to recognize binding motifs irrespective of where they were in the sequence vector. To achieve this, the training set was expanded to include each sequence in 6 different registers within an extended sequence vector (see Methods). All shifted registered versions have the same target binding values associated with them for training. As can be seen in the fourth column of Table 2, this generalization greatly improves the predicted ranks of almost all of the mAbs cognate epitope sequences such that 5 rank in the top 0.001% of sequences and 7 are in the top 0.01%. The one exception is 9E10.3, which ranks significantly lower with shifting than without.

It is worth considering why this works as well as it does for most of the mAbs. The effect on the prediction for Ab8 is particularly significant, improving its ranking by 4 orders of magnitude. It is likely this is due to the fact that there are so few sequences which contain the needed sequence-specific information. The matrix manipulations in a neural network are register-specific; an amino acid in the first position is always treated the same

way mathematically. Thus, with sequence shifting, the number of available sequences to train on is effectively increased 6-fold. Further, any bias that may exist in amino acid composition of a particular position or any effects from the fact that the sequences are not all the same length are effectively decreased. This means of improving the predictive power of the model could be applied to other situations, such as whole sera for epitope predictions associated with an immune response to disease[7]. While similar in some respects to using a convolutional neural network, it differs in that the entire sequence is considered at once, rather than small pieces of it.

### Training after normalizing for the level of composition-dependent binding

In any antibody interaction with a target sequence, there will be some level of interaction which is strictly sequence dependent (sequence-specific) and some level of binding that depends only on the overall nature of the peptide (charge, hydrophobicity, etc.). One way to separate out the binding that is strictly dependent on sequence order is to determine what fraction of the binding for each peptide can be determined only from information about the peptide composition and then subtract this. Supplementary Fig. S5 shows the result of fitting the binding of the mAbs to the peptide array by only considering 16 linear coefficients, one for each amino acid used, that operate on a composition vector representing the peptide (a vector made up of the number of each amino acid present). The part of the $\log_{10}$ binding that could be described by composition alone was subtracted from total $\log_{10}$ binding and the neural network was then trained on the resulting sequence dependent binding for each mAb. Using this approach, there is very little change in most of the mAbs. For Ab8, there is a modest improvement and the 9E10.3 cognate sequence moves up in rank significantly. Apparently, removing composition-dependent binding from Ab8 and 9E10.3 allows the neural network to focus on learning the sequence dependence without simultaneously modeling general compositional binding. The final ranks obtained for all mAbs using the full model are within the top 0.01% of the 1,000,000 random sequences and 7 of the 8 are within, or very near, the top 0.001%. Supplementary Table S2 gives the predicted top 20 random sequences for each mAb. One can see the relationship to the cognate sequences in each case for at least some of the top sequences. In the case of Ab8 in particular, there are also apparently unrelated sequences that contain multiple arginine and tyrosine residues. As a control, the model was also trained after randomizing the order of the sequences on the array relative to their binding values. This gives ranks near 50% after averaging, as one would expect (Supplementary Table S3).

### Exploring different sized sections of the antigen sequence around the cognate sequence

Choosing 7 amino acids as the length of each cognate epitope is largely arbitrary. While past work has identified the region of the antigen that is important in binding, the length of the region that influences that binding is not obvious. Supplementary Tables S4 and S5 shows the results of using sequences of different lengths from 6 to 10 residues taken from the cognate epitope region of the antigen to each mAb. Supplementary Table S4 shows the sequences used and Supplementary Table S5 shows the rank using the full model (weighting, shifting and removing compositional binding) of each of those sequences among one million randomly generated sequences of the same length. The changes in rank are relatively small for different sized regions of antigen sequence used, with some mAbs slightly favoring longer sequences and some slightly favoring shorter ones.

### Predicting which amino acids are most important in the cognate sequences

The neural network models can also be used to predict the binding of all possible single amino acid substitutions of the mAb cognate sequences using the 16 amino acids that make up the array peptides. The linear binding at 2 nM concentration was predicted and normalized to the cognate sequence value for each possible single amino acid mutation (Supplementary Fig. S6). For four of the mAbs, these substituted sequences were also synthesized on a

separate array and their values measured for comparison. The predicted and measured values for those four mAbs are shown in Fig. 2. The color scheme shown is limited to the range between no binding (0) and 1.1 times the cognate binding to clearly show binding at or below the cognate level. The correlation between the measured and predicted values using all of the values in each matrix is also shown. The agreement between predicted and measured variants of the cognate sequences is generally good, with the model accurately determining which amino acids at which positions are critical in most cases except in a few cases. For Ab8, W is predicted to be easily substituted by several other possible amino acids. However, in the measured peptides W can only be substituted by Y, and that with some loss of binding. In TF3B5, E is predicted to be essential, but in the measured sequences, it is clear that E can be replaced by any of five other amino acids with no loss in binding. In 9E10.3, the ED pair towards the end of the cognate sequence shows somewhat different variability of substitution between predicted and measured values. This cognate sequence includes three similar amino acids in a row (EED) and it may have been hard to differentiate the effects of the last two. However, in general, the prediction of which amino acids are most essential is in line with what is measured.

### Mapping mAb binding to antigen sequences

The ability to apply the neural network binding models developed here to biological sequences is currently limited by the fact that 1) only 16 of the 20 natural amino acids are included in the peptide sequences on the arrays used in this study and 2) three dimensional structures are not currently considered in determining binding sites in the application of the models onto proteins which limits the ability to map non-continuous epitopes. However, to demonstrate proof of principle with the current model system, Fig. 3 shows color maps of the binding of each of the mAbs modeled here to its respective antigen structure. The 4 missing amino acids were dealt with by similarity substitution (I→V, T→S, M→L, C→S) determined using the PAM250 similarity matrix[31], based on comparison of various similarity matrices[32]. The sequences of the appropriate antigen subunit are rendered as Alphfold2[33] predicted structures to provide an entire representation of the sequence, including unstructured regions not present in the crystal structures (Table 1 gives the Uniprot ID numbers from which the structures were derived). The sequences of the antigen subunit were tiled as overlapping 7 residue peptides (6 residue overlap). The 2 nM binding concentration was then predicted and each tiled peptide assigned a color between green and red based on the $\log_{10}$ predicted binding level. The log was used here to allow visualization of very weak sites. One can see one such site, for example, as a brown region at the C-terminus of P53 when Ab8 binding is predicted.

In most of the mAb binding maps, there is little significant binding other than the cognate sequence (bright red). However, 9E10.3, in contrast to the others, shows three strong binding sites. The issue in this case is not an inaccurate prediction of the model but rather a combination of there being three similar sequences within the myc antigen and the amino acid limitations of the model system used. The three sequences of the myc protein antigen that bind strongly to 9E10.3 are: KLISEE (cognate, only this particular helix was used to select the mAb, see product literature, ThermoFisher, Cat# MA5-12080), KLVSE and MVTE. However, because the model system contains only 16 of the 20 amino acids, it was necessary to replace I and M with V and L, respectively. Thus, the sequences become KLVSEE, KLVSE and LVSE. Looking at the substitution matrices of Fig. 2, one can see that the critical part of the cognate sequence is LVSE. Development of arrays incorporating all amino acids will be needed for their more general use in identifying epitopes within proteins and proteomes.

### Robustness of the trained models

As shown in Table 2 and Supplementary Table S5, the models, when trained on the entire array of about 122,000 peptide sequences, correctly recognize the cognate sequences for each of the mAbs as being a top binding sequence in a large library of random sequences. How robust are these models? One way to explore that is by decreasing the number of peptide sequences used in the training. Supplementary Fig. S7 shows the result of decreasing the

# Mutation Matrices

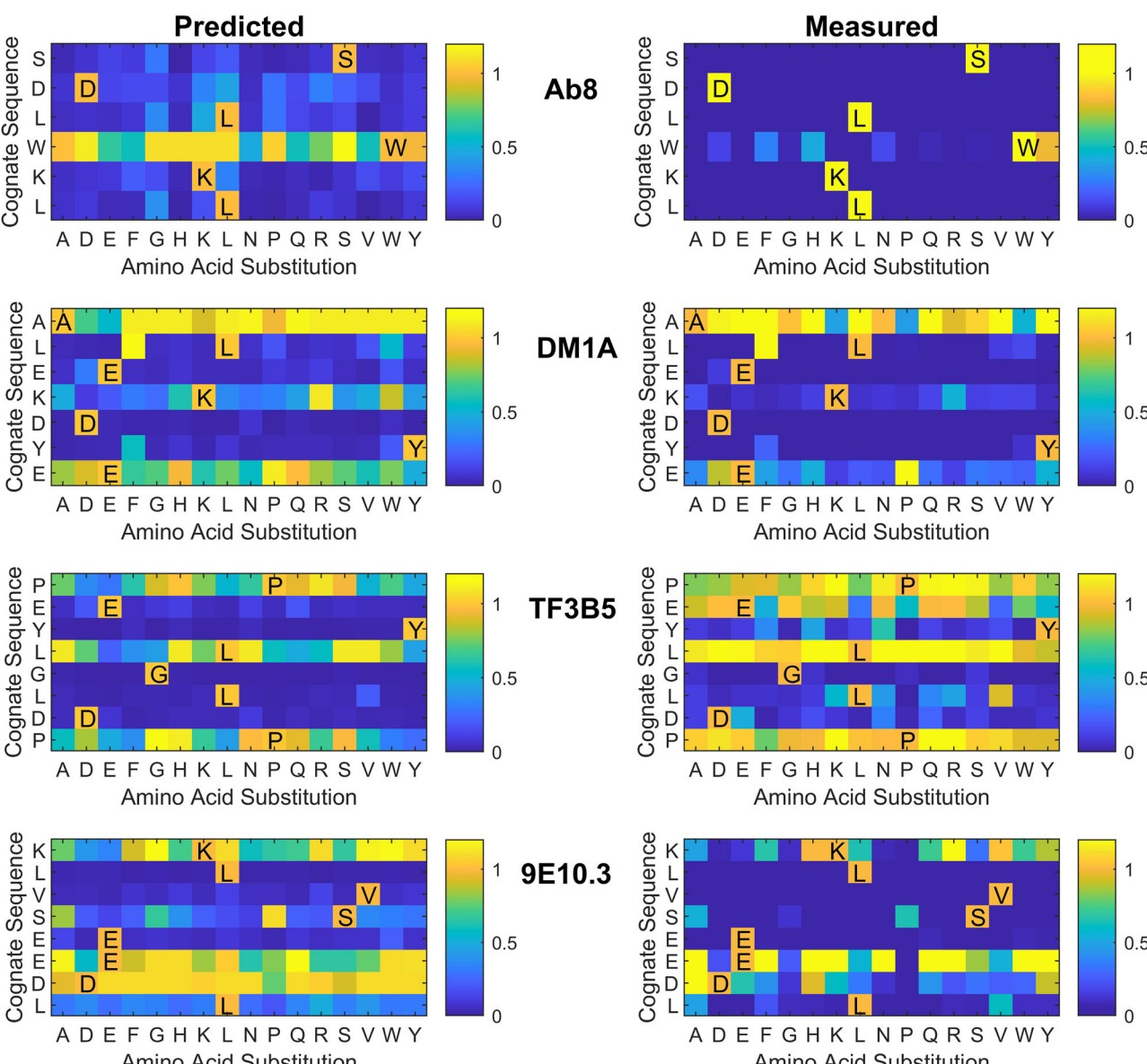

**Fig. 2 | Single amino acid mutation/substitution matrix for each of the mAb cognate sequences.** The y-axis is the cognate sequence, and the x-axis is the amino acid used for substitution. The color is proportional to the binding predicted by the full neural network model (left side) and the measured binding for the cognate sequence and all single amino acid variants (right side). Note that the measured binding values come from a different set of arrays with different sequences and layout than the arrays used for the bulk of the work and thus what are presented for both predictions and measurements are relative values rather than absolute binding. The 2 nM concentration mAb data was used for the predicted values shown. Note that any sequence with a value of 1.1 times the binding of the cognate sequence or higher is shown as yellow. This allowed better resolution of values less than 1.

fraction of the sequences used in the training stepwise from 100% (~122,000) to 6% (~7300). By far the most fragile model is that describing Ab8 which gives a near random result when 75% of the peptide sequences are retained. (Note that this is spuratic; some reduced models are almost indistinguishable from the model using all sequences and some are dramatically worse). The rest are stable at least to the point where 50% of the sequences are used in training. Ab1 and 9E10.3 drop in rank at 25% of the original number of sequences, but the other mAb models decay more gradually between 50% and 6%, with the exception of TF3B5 which seems to predict very well even with only 6% of the original array data to train from.

Another way to explore the robustness of the models is by specifically removing sequences that are similar to the cognate sequence. To accomplish this, each sequence on the peptide array was compared to the 7mer cognate sequence of Table 1 in all possible registers, even those that only partially overlap, and training sets with only peptide sequences with no more than 6, 5, 4, 3 or 2 amino acids that aligned were retained and new models were generated based on these (these models were trained with weighting, shifting and removal of compositional binding, see Table 2). Supplementary Table S6 shows how many sequences were removed from each of the training sets for each of the mAbs, and Supplementary Table S7 shows the numerical ranks for each model. The results are graphically presented in Fig. 4. As was seen when decreasing the total number of peptide sequences used in training, Ab8 proved to be the most fragile model requiring sequences with at least 5 amino acids in common with the cognate epitope to maintain accurate prediction. The Ab1 cognate is predicted well as long as at least 4 amino acids are allowed to be in common with its cognate sequence.

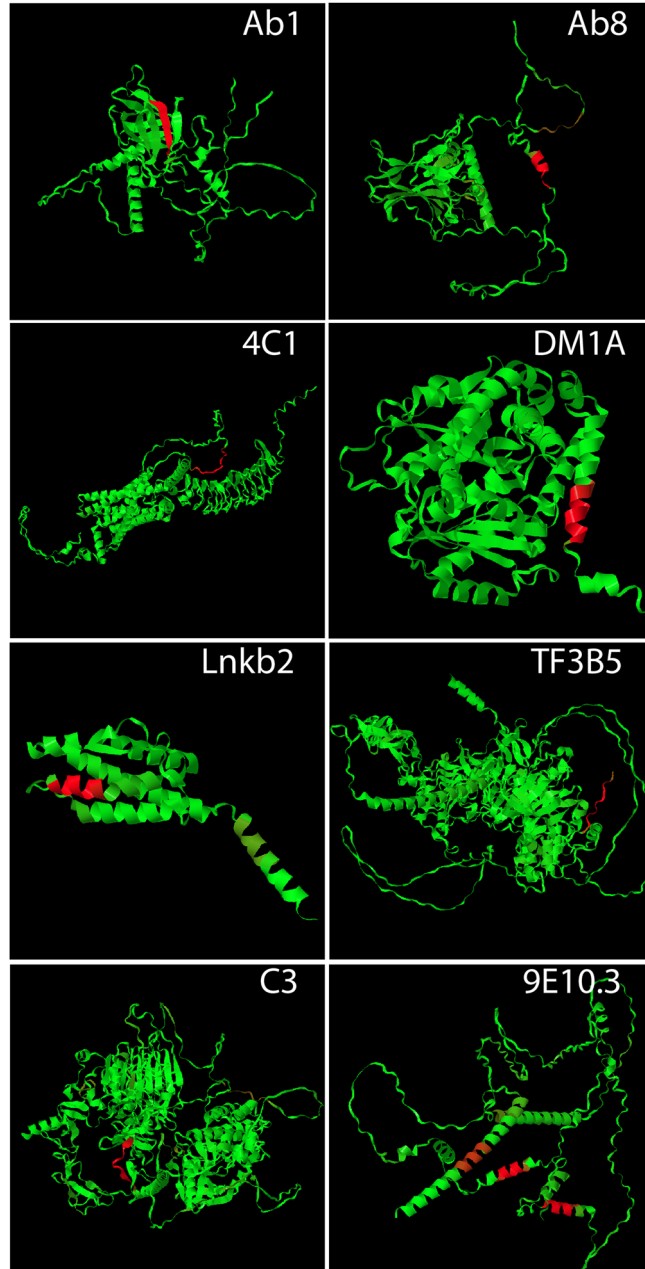

**Fig. 3 | The sequence of amino acids was tiled into overlapping peptides of 7 residues (6 residue overlap).** The $\log_{10}$ binding values at 2 nM concentration were predicted and converted to a color, increasing from green to red, and mapped onto the Alphafold2 representation of the sequence provided on the Uniprot website (Uniprot IDs given in Table 1).

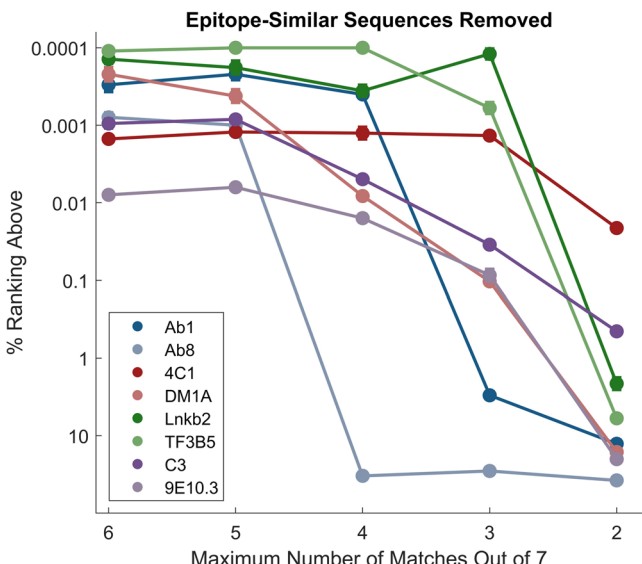

**Fig. 4 | Ranks of the 7mer cognate epitope (Table 1) in 1 million random sequences using training sets of array peptides in which sequences similar to the cognate sequence have been removed.** The Y axis is the percent rank, with 0.0001% signifying rank 1 in a million. The X axis is the maximum number of amino acids that were in common with the epitope in the peptide array sequences used for training. The points and error bars (standard error of the mean) represent the average and standard error of 5 randomly chosen sets of 1 million comparison peptides each and 12 randomly initiated models. In most cases, the error bar is too small to see in the plot. See Supplementary Tables S6 and S7 for tabular values.

The others drop gradually, but even when only 3 common amino acids are allowed, their cognate sequences are still ranked in the top 0.1% of the random sequences, and three still rank in the top 0.001%. Clearly there is a range in the robustness of the trained models that depends on the details of the data available and the nature of the binding interaction.

## Conclusions

The goal of this work is to explore what can be learned about the binding of a target molecule that is highly specific for a particular amino acid sequence by applying machine learning to a dataset derived from binding the target to a sparse, unbiased sample of short amino acid sequences. Here, mAbs with specific linear epitopes were chosen both as an example of a sequence specific target molecule and due to the inherent interest in understanding

and predicting mAb binding. Past work from this lab has demonstrated the ability to use similar datasets and algorithms to predict binding of peptides to the surface of multiple proteins and for total binding of serum IgG to peptides, but recognizing the very specific binding of mAbs to their linear epitope sequences represents a more difficult challenge. As shown in Table 2 and Supplementary Table S5, models can be developed which do recognize the cognate sequences as being highly specific based on the binding of the mAb to a sparse, unbiased sample of sequences. Further, Fig. 2 demonstrates that the models can differentiate between mAb binding to their cognate sequences and to sequences that differ by only one amino acid, making it possible to map, with reasonable accuracy, the relative importance of different amino acids in mAb binding.

The neural network approach used was quite simple: a fully connected network with a one-hot representation of each amino acid sequence as input and $\log_{10}$ binding of that sequence to four different mAb concentrations as the target for optimization. However, in order for the models to predict well for all of the mAbs, additional modifications of both the input and target data were required. The neural network model in its simplest form was modestly predictive for most of the mAbs with the exception of Ab1 and Ab8. One issue with binding of mAbs to random sequences is that the vast majority of the peptides show essentially baseline binding. Weighting the peptides that bind more strongly improved the model for most of the mAbs. Forcing the neural network to ignore the absolute position of the sequence in the vector and generalize its learning so that it recognized the sequence regardless of its position within the artificial register of the input vector helped further in most cases. This form of data augmentation by training on multiple sequence registers is well-established to help generalize convolutional neural networks for image recognition tasks[34]. It is surprising in many respects that the much simpler neural network model used here (2 hidden layers of 250 nodes, fully connected) would be able to utilize this data augmentation to generalize in this way. The one mAb for which training on multiple sequence registers resulted in poorer recognition of the cognate sequence was 9E10.3, but by removing binding due to amino acid composition, the cognate sequence for this mAb was predicted in the top 100

sequences out of one million, presumably by focusing the learning of the neural network on the sequence specific aspect of binding.

Note that because there was so little high binding data available to learn from in some mAb binding datasets, the absolute predicted values of the cognate sequences were often underestimated (Supplementary Fig. S4). This was particularly true for Ab1 and Ab8, where high binding sequences were extremely scarce. Apparently, it was difficult for the model to learn to predict the absolute binding values of sequences that were more than an order of magnitude higher than anything the model was trained on. However, the relative values, and thus the rank of the binding value, was predictable, even when the absolute value was underestimated. From a practical perspective this is the more important outcome.

While predictive when all peptides on the array were used in the training, the models for Ab1, and particularly Ab8, were not robust to removal of sequences. For Ab8, removing just 7 of the most similar sequences to the cognate sequence rendered the model effectively without any predictive power (Fig. 4, Supplementary Tables S6 and S7). Again, this is consistent with the relative scarcity of sequences on the array that bound substantially to these two mAbs.

This brings up a valid question about the extent to which the model is generally learning the landscape of mAb binding affinity as opposed to simply memorizing a few closely related high binding sequences on the array. For Ab8 in particular, where the model was so dependent on so few sequences, this may indeed be partially the case, though the model still needs to recognize the sequence in the context of other sequence around it. However, for most of the mAbs there are several arguments supporting the conclusion that the model learns to put multiple parts of the sequence-binding information together in recognizing the cognate sequence as highly specific. First all but two of the mAb cognate epitopes rank in the top 0.1% of all sequences even when we limit similarity of training sequences to only three out of seven residues (Fig. 4, Supplementary Tables S6 and S7). Indeed, even when limited to no more than 2 amino acids in common with the cognate sequence, half of the mAb cognate sequences are ranked in the top few percent. Second, the ability to recognize that some single amino acid changes can completely block binding while others maintain or increase binding (Fig. 2), also suggests that the model has a more comprehensive view of sequence dependence than just remembering one or two examples. Third, the ability to work with different lengths of cognate sequence up to 10 (Supplementary Tables S4 and S5) is consistent with learning sequence context and what to include and what to ignore. Finally, the fact that the model generally predicts the binding values of peptides on the array that were held out of the training and used as a test set, regardless of their sequence, supports the idea that the learning is general, rather than simply memorization (Supplementary Fig. S4, in cases where there is a significant dynamic range of binding, it is well predicted).

As described in the introduction, there are a number of different approaches to predicting epitope sequences, as well as off-target binding sequences, of mAbs. This includes peptide array approaches in which proteins or proteomes of interest have been directly tiled[15,16]. It is fair to ask, why wouldn't one simply tile the proteome of interest rather than generating model predictions from sequences broadly sampled from all possible sequences? There are two reasons for exploring the approach described here, one practical and one more fundamental.

From a practical perspective, the array-based approach employed here and the comprehensive model it produces provide both a universal assay platform that can be applied to any mAb at low cost/high throughput with complete flexibility in the evaluation of mAb binding to any sequence. For example, suppose one is interested in exploring all potential targets (or off-target binding) in the human proteome for 100 lead mAbs. A single 8 inch silicon wafer substrate used in array generation contains >300 individual arrays, thus the arrays from one wafer could assay all of these mAbs at three concentrations and create models for each (and one photolithographic fabrication station generates 4 wafers at a time). In contrast, tiling the ~11 million overlapping peptides in the human proteome would be very difficult and expensive to do at this throughput. In addition, such a tiled array would

only be generated for a particular reference proteome and would not necessarily contain mutations of interest in a particular disease state or for individual proteomes. Using a comprehensive model based on much smaller, near-random sequence arrays, one can predict mAb binding to any sequence including the proteome sequences of individuals determined from their personal DNA sequences. Models also allow the prediction of binding to sequences that may be challenging to synthesize in a tiled library. Of course, the current system is limited in this regard, both because it focuses on linear epitopes and because only 16 of 20 amino acids were used, but it provides a proof of principle upon which more useful systems could be developed.

As described above, the focus of this work is more fundamental, and from that perspective, the study's outcome, building on previous work from this lab[6–8], is important for another reason. Almost all of the machine learning work to date that relates amino acid sequence to structure and function has involved taking a set of known functional sequences (folded proteins, antibiotic peptides, epitopes, etc.) and learning from known examples. Alphafold2 is perhaps the most ambitious such achievement[33], taking ~140,000 known examples of protein structures and generalizing to predict the structure of any sequence. In such cases, one does not know how much of the primary sequence space the predictions are actually applicable to because the training is biased towards known function and there is usually no way to empirically test all the possible sequences to see how significant that bias is. However, in the work described here, the training was done without any known examples, based on a largely unbiased sampling of the possible sequences. Yet this very sparse and unbiased sample of a trillion-member sequence space is sufficient to provide a reasonably accurate model of the entire molecular recognition landscape, including predicting the relative binding of something as specific as a mAb cognate epitope. Thus, at least for a relatively modest sized sequence space in the context of molecular recognition, it is possible to create a comprehensive model from sparse sampling where there was no a priori bias in the sequences used to train the model and no reason to expect bias in its predictions.

## Methods
### Peptide arrays
Peptide microarrays containing diverse peptides were synthesized at Cowper Sciences, Inc., (Chandler, AZ) following a previously described manufacturing process[35]. In brief, 200 mm silicon oxide wafers were functionalized with an amino silane coating and terminated with Boc-Glycine. Each peptide sequence was constructed on the functionalized surface by repetitive photolithographic deprotection and coupling cycles with preset mask-amino acid combinations. A total of 16 amino acids were used (ADEFGHKLNPQRSVWY) to generate a library of 126,051 unique peptides with a median length of 9 residues and range from 5 to 13 amino acids. This arrangement provided coverage of 99.9% of all possible 4-mers and 48.3% of all possible 5-mers consisting of the 16 different amino acids used. Only the 121,715 peptides between 5 and 11 amino acids in length were used in this study. Synthesis verification was performed by MALDI mass spectrometry (see supplementary information).

### Monoclonal antibody sample preparation
All aliquoting and dilution steps were performed using a BRAVO automated pipetting station (Agilent, Santa Clara, CA). Eight individual murine monoclonal Antibody standards used in the study (referenced in Table 1) were initially prepared at 100 nM and 10 nM stock concentrations in 1% mannitol/PBST buffer. Dose curves from each Ab were prepared by dilution at 4x steps from an 8 nM starting concentration. All mAb concentrations were assayed in quadruplicate.

### Antibody assays
Peptide microarray slides were assembled into standard 96-well format cassettes (4 microarray slides per cassette). An automated assay workflow was initiated with slide rehydration, by incubating a cassette in 1xPBST for 20 min at 57 °C. Buffer was removed from the cassette and 90 uL of each

prepared monoclonal antibody sample was applied to assigned arrays. The cassettes were then covered with aluminum seal and incubated for 1 h at 37 °C while mixing on TeleShake95. Following incubation, each cassette was washed with 10xPBST (300uL/wash/well) using a microtiter plate washer (BioTek Instruments, Inc., Winooski, VT). Labeled anti-Murine anti-IgG Antibody conjugated to AlexaFluor 555+ (Invitrogen Thermo Fisher Scientific) was used as secondary antibody, at 4.5 nM concentration in 0.5% casein/PBST buffer for 1 h with mixing on a TeleShake95, at 37 °C. Following incubation of the secondary antibody, the slides were again washed with 10xPBST, followed by 3x washes with water. The cassette was disassembled, and slides were sprayed with isopropanol and spun dry.

Dried slides were imaged using an ImageXpress imaging system (Molecular Devices, San Jose, CA). Post scanning, flat-field correction was applied to generate a single array TIFF image file. Image analysis was performed using custom software that enables automatic grid placement and feature intensity extraction. A murine sera control sample was run on each slide and used to assess slide-to-slide reproducibility, with an accepted replicate correlation coefficient of >0.95.

### Neural network architecture

The basic neural network architecture has been described previously and schematics provided[6,7]. A fully connected neural network was used and the inputs were one-hot representations of sequences. Various network training parameters were optimized (numbers of hidden layers, numbers of nodes, epochs trained, minibatch size, etc.) and a network with 2 hidden layers and 250 nodes was selected (full details are contained in the code made available on https://zenodo.org/records/10262899). Training was performed relating the sequences to binding values on the array. All of the measured sequence-binding pairs in the 5–11 amino acid length range, except the sequences on the array that were cognate epitopes of the 8 mAbs, were used in the training. This did not introduce bias, since neither the cognate sequences nor any of the million random sequences later used in predicting the rank of the cognate sequences were part of the training. (For completeness, Supplementary Fig. S4 shows a 10-fold cross validation of predictions of the array data.) Each mAb was separately modeled, but all four concentrations were modeled at once as a four-column target. What is different in the analysis of this data from previous approaches from this lab[6,7] is the way the input sequences and the targets were modified prior to training the neural network.

### Padding and designation of N- and C-termini

The peptides on the array used for this analysis had a distribution of lengths from 5 to 11 residues, centered near 9 residues. To deal with varying lengths, sequences less than 11 residues were padded with token "X", the N-terminus was marked with token "(" and the C-terminus marked with token ")". As a result, the sequences that were assigned one-hot representations had 19 different one-hot bits, 16 for the amino acids used in the synthesis and 3 for the "X()" token. For example:

    (PWRGPWARV)XX
    (LPGVQG)XXXXX
    (GNFAYQRDG)XX

This approach to designating the sequences becomes particularly important when the position of the sequences within a larger character vector is varied.

### Weighting of high binding data

As described in the text, the number of sequences that bound to mAbs with values substantially greater than background was often small. As a result, the background sequence-binding pairs often completely dominated the loss function in the training and key binding information was not emphasized. To avoid this, multiple copies of higher binding sequence-binding pairs were used. The sequences were binned by their binding values at a concentration of 8 nM (bin width of 0.2, binning the $\log_{10}$ binding values) and any bin with less than 300 values in it was expanded to 300 values by randomly picking sequence-binding pairs within the bin and copying them.

No one sequence-binding pair was allowed to be copied more than 100 times.

### Creating register shifted versions of the input vectors

One of the most effective approaches used to improve the ability of the neural network to predict high binding of mAb cognate sequences was by forcing the network to consider multiple frames of each sequence as being equivalent. Each sequence-binding pair in the training set was expanded to 6 copies, each in a different frame within the original character vector as follows:

    (PWRGPWARV)XXXXXX
    X(PWRGPWARV)XXXXX
    XX(PWRGPWARV)XXXXX
    XXX(PWRGPWARV)XXXX
    XXXX(PWRGPWARV)XXX
    XXXXX(PWRGPWARV)XX

In the training, the 6 sequences are all assigned the same measured binding value. This greatly improved the fidelity of the model in most cases. Evaluation of the model was performed in the same way. In other words, any sequence for which a binding value was to be predicted was also submitted as 6 copies in 6 frames as above and in this case, the highest binding value was then chosen for ranking that sequence. Since all sequences were processed the same way (in this case all million randomly generated sequences), this does not introduce any bias in the overall ranking.

### Compositional binding estimates

Estimates of compositional binding for each sequence were determined by taking all sequences in the training set and assigning a simple vector of 16 composition values (the number of occurrences of each of the 16 amino acids used in that sequence). The values were then used in a linear fit (16 coefficients and a bias term). In the case of sequences to be predicted, the compositional binding was determined by applying the linear model coefficients learned from the fit of the training set.

### Statistics and reproducibility

The binding values of each array peptide for each of the 8 mAbs and each of the 4 concentrations were measured 4 times and averaged. No values were excluded. The sequence-binding models trained on the binding of all four concentrations to each sequence was applied to predict the binding at the four concentrations to each of 1 million randomly generated sequences as well as the cognate sequence for that mAb. The binding of the million random sequences and the cognate were ranked separately at all four concentrations. For each sequence, the highest rank value (closest to 1) was selected and these values re-ranked to remove any duplicate ranks. The major variation in the resulting rank comes from three sources: errors in the measurements, the fact that the neural network starts with a set of randomly selected weights, and the fact that the cognate epitope was ranked in a set of randomly chosen sequences. To determine this variation and the resulting error of the mean in Table 2, 12 neural network models (starting with 12 sets of random weights) were determined for each mAb. These 12 models were used to predict the binding to each of two independent sets of 1 million random sequences and the results of the 12 models for each of the 2 sequence sets averaged. The entire process was then repeated with new random seeds 5 times to generate the error statistics shown. This should take into account the possible sources of variance.

### Data availability

The array data for all 8 mAbs is provided on Zenodo (https://zenodo.org/records/12510566)[36] both with all 4 replicate binding measurements averaged at each concentration and as four unaveraged replicate files. Note that Fig. 1 as made from the data in the averaged array datafile. Also, excel files are uploaded that contain the data in Figs. 2 and 3 (labeled as Fig. 2_Data.xlsx and Fig. 3_Data.xlsx). The data for Fig. 4 is provided in the Supplementary material pdf as a table.

## Code availability

Matlab code needed to train the neural network model and determine the ranks of the cognate mAb epitopes in 1 million randomly generated sequences has been deposited in the same location as the datasets (https://zenodo.org/records/12510566)[36]. The Matlab script contains all needed functions and if it is run in the same folder any of the data files (averaged or individual replicates) it will generate the ranks of the mAbs within the 1 million randomly generated sequences, regenerating results similar to Table 2 but without as much averaging of different trained models. A detailed description is given in the Supplementary Materials text and Supplementary Table S8. The code was developed on Matlab 2022a, but runs on 2021 and 2023 versions as well.

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

## Acknowledgements
This work received support from Arizona State University and in-kind support from Cowper Sciences.

## Author contributions
P.B. and A.S. performed the original curation and algorithm development during the early part of the project. L.K. was involved in collection and curation of the final dataset, performed aspects of the final analysis and helped design the final algorithms. O.T. was involved in assay design and

execution. A.T.T. provided insight into machine learning methods and helped formulate the analysis. N.W.W. managed the project, performed the final coding and calculations associated with the neural network analysis and wrote the first draft of the manuscript.

## Competing interests

The authors declare the following competing interests: L.K. and O.T. received renumeration from Cowper Sciences during this time period either as a consultant or employee. Cowper may benefit from the publication of this manuscript. L.K. is the founder of Biomorph Technologies, LLC which may benefit from the publication of this manuscript. A.S. worked for Prellis Biologics, Inc. during the time this work was being performed and her company could benefit from its publication. A.T.T. worked for iBio Inc. during this time period as an employee. iBio may benefit from publication of this manuscript. L.K., P.B., A.S. and N.W.W. were co-inventors on an invention filed as a provisional patent during this period, "Methods and Related Aspects for Predicting Antibody Binding", provisional patent application 65/593,187. A PCT has not yet been filed. N.W.W. was an unpaid and unofficial advisor to Cowper Sciences, Inc. during the period of this work.
