## [Peer review file · Communications Biology]

Reviewers' comments:

Reviewer #1 (Remarks to the Author):

In this paper a combined in-silico experimental pipeline is presented to map the linear epitope space relating to a particular antibody. I think it is a valuable study given that the authors can address the following comments:

- It is not clear to me whether antibody sequence information is taken into account. From my reading, each antibody model is trained separately on the linear epitope data. However there are also mentions of 'binding pairs' in the manuscript.
- I would emphasize the point that authors went 'the other way' to traditional methods that map abs with respect to an antigen. Mapping different antibodies with respect to an antibody has a clear discovery purpose as it gives novelty and priority to a typically known target. By contrast mapping epitopes to a known ab has a seemingly tenuous application. Chief potential application that is only scarcely mentioned is off-target binding as it is a big problem in contemporary immunotherapies. Authors could expand upon that.
- Are there similar antibodies in the PDB to the ones mapped here? If so are their linear epitopes (or subsets of discontinuous) within the predicted sets?
- There are many linear epitope prediction softwares (e.g. <http://tools.iedb.org/main/bcell/>) I think it would be worth it to either perform benchmarking or say why it does not apply here.

Reviewer #2 (Remarks to the Author):

In this work, Bisarad and colleagues investigate antibody binding against random peptide arrays. This work is very interesting and rigorously carried out. I have a few comments.

- have you considered data leakage in your model? That means, have you tried splitting the training/test data by sequence similarity? To quantify generalization of the predictor?
- Are there peptides that are very similar in sequence yet have different binding? Are these predicted correctly? Can you make a plot where you plot delta peptide sequence

similarity vs delta accuracy? or something like that?

- have you tried baseline models which just take into account amino acid composition but not the sequence?
- Are there differences prediction performance between the antibodies? if so, why?
- Can you add an overview Fig 1 of the experimental/computational setup?
- Did you perform controls (randomize peptide sequence  does prediction accuracy go to random?)
- Please provide numbers/percentages in figure 1 as to how many peptide are above noise.

Reviewer #3 (Remarks to the Author):

Summary

Bisarad, Kelbauskas, Singh et al. present a dataset of eight antibodies with binding data against 122k randomly designed peptides. They show that, for each antibody, a simple Multilayer Perceptron (MLP) trained on this data consistently ranks the original (cognate) linear epitope highly among 1m different random peptides of the same length. The authors hypothesise this high throughput screening and ML model could be useful for selecting highly specific mAbs for therapeutics or diagnostics.

The paper is clearly written but lacks a strong argument linking the method results (high ranking of the cognate epitope) with the original hypothesis, and the use case and impact of the work is not clear. The paper also lacks robust analyses in places. We have highlighted some of our main concerns below.

Major

1. The paper does not address the question of whether or not the high throughput screening and ML model can identify polyspecific binding. Instead, the paper focuses on the ability of the trained MLP to rank the cognate epitope highly among a random background. This result is not clearly linked to the original question. Though many random peptides were also ranked highly by their model (above the cognate epitope), the authors did not test their binding experimentally.

2. In Figure 3 (page 11) the authors consider the entire protein antigen of each mAb, break it down into peptide chunks, and rank each chunk using their trained MLP. Encouragingly, the cognate epitope ranks highly. However, if the antigen were known a priori (as is often the case) and the goal was simply to identify any linear epitope, then would the more efficient experiment not be to select the e.g. 200 or 1k peptides that

comprise the antigen and test for binding against these instead of 122k random epitopes?

3. Assuming the goal of the paper is actually focused on ranking the cognate epitope highly (this should be justified), then more robust statistics should be provided here. Given final comparisons are only made to peptides of the same length as the cognate epitope (6-7 amino acids) and only 16 amino acid types are allowed, the total search space is 16^6 or 16^7 (~17m-268m). If the cognate epitope were not known a priori, the chances of selecting the cognate epitope in each instance given 1m designs is 6% or 0.4%. To be robust, the authors should evaluate how highly the cognate epitope ranks amongst the entire possible sequence space - this should certainly be computationally possible for length-six designs at least. If however, the emphasis remains on predicting poly-specificity, with ranking of the cognate epitope being a measure of evidence for this, then this should be more clearly emphasised and argued for.

4. However, to be yet more robust, it should be assumed that the cognate epitope length is also not known a priori (peptides of length 5-11 are used for training). In this instance, the cognate epitope should be evaluated against designs of random lengths. Here, the sequence space expands to ~1 trillion (infeasible to exhaustively search), as stated in the paper (page 2), and the probability of selecting the cognate epitope by chance given 1m random designs is just 0.0001%. Again, it should be considered whether or not selecting the cognate epitope in the top X designs is the true goal of the paper.

5. Only one MLP model is considered for training. It would be useful to try alternative architectures such as a random forest or convolutional neural network (CNNs) for comparison. CNNs would be particularly suited to the task due to their ability to pick up position-independent motifs. In addition, the authors do not describe if/how they optimised their MLP architecture or training hyperparameters. The number of training epochs or iterations is also not stated.

6. The significance of the steps taken to improve the MLP performance - oversampling positive sequences given the large class imbalance, and "shifting" the sequences - are overstated. The paper says "The biggest advance in the model came from forcing the neural network to ignore the absolute position of the sequence in the vector and generalize its learning so that it recognized the sequence regardless of its position within the artificial register of the input vector. This form of data augmentation by training on translational variants is well-established to help generalize convolutional neural networks for image recognition tasks. It is remarkable in many respects that the much simpler neural network model used here (2 hidden layers of 250 nodes, fully

connected) would be able to utilize this data augmentation to generalize in this way." These training techniques are standard, necessary, and not as remarkable as claimed.

7. Given the large training set size (122k), lack of sequence identity filtering, proven sequence similarity of the top hits (Figure 1), and oversampling of these top hits, the MLP appears to be simply memorising its input and struggling to predict anything out of distribution. It would be useful to repeat the training with strict sequence similarity cut-offs in the train and validation datasets - this would help justify the claimed benefits of the sparse sampling strategy. Additionally, models with decreasing training set sizes could be trained to determine the minimum training set size to still result in a high ranking of the cognate epitope.

8. Why were the cognate epitopes of TF3B5, C3, and 9E10.3 not included in the 122k peptides tested for binding? There must have been space on the array and their exclusion is not explained, simply stated.

9. Correlation coefficients and the gradients of the lines of best fit should be added to Figure S4 to better understand how well the predicted and measured log₁₀ values agree.

10. The language used to describe neural network training and architecture could be improved for clarity and to remove unnecessary complexity, e.g.:

- "The matrix manipulations in a neural network are register-specific; an amino acid in the first position always "sees" the same set of processes take place as the information moves through the neural network."
- "it recognized the sequences regardless of its position within the artificial register of the input vector."

Minor

1. The Introduction states "Binding typically involves interactions between the mAb and 5-8 amino acids and in a specific spatial arrangement". However, for full-length proteins, which this paper considers, the full epitope is often composed of multiple linear/conformational epitopes, typically totalling 10-20 residues (Reis et al., 2022). This fact should be clarified here.

2. Figure 3's image quality is very low and should be improved.

3. Ideally, should additional epitope regions be identified in Figure 3 that are close in space to the cognate linear epitope (if they exist)?

4. Given the greater interpretability and similar performance, why was the concentration maximum (Table S1) not used in place of the slope value (Table 2)?

5. Can the similarity between the predicted and measured values in Figure 2 be quantified in some way?

6. In the methods section “since neither the cognate sequences or any of the million random sequences ... were part of the training” should be amended to ‘since neither the cognate ... were in the training set’.

Response to Reviewer Comments

Predicting Monoclonal Antibody Binding Sequences from a Sparse Sampling of All Possible Sequences

Bisarad et al.

COMMSBIO-23-4495-T

General Comments

This was a helpful set of reviews both in assisting us in seeing how to better crystalize a set of concepts that are hard to portray and in recognizing ways to make the manuscript stronger overall.

There are two general themes that run through the comments from all reviewers. We will address these first and then discuss each of their comments individually and how we responded to them.

The goal of the work. The first issue raised by all reviewers in various ways has to do with what the goal of the paper is. Admittedly the original version of the manuscript was somewhat unclear in this regard. The hypothesis that we are testing is that one can build a sequence-binding relationship for a large combinatorial space of amino acid sequences starting with data from a very sparse and essentially random sample of that sequence space. The fact that we are using mAbs as the target in binding is less because of their potential utility than it is because they represent the ultimate test of the ability to recognize a very specific sequence interaction. We now try to put this in context more clearly both in the Abstract and in the introduction on page 3. We also now describe this work in the context of our larger research efforts on page 1. This is the fourth paper in a series. The first paper used a set of common proteins as targets and showed that we could develop a sequence binding relationship from sparse sampling of a random combinatorial space that accurately described the binding of that protein to sequences not used to train the model. However, this kind of binding is not very specific and that makes it easier to predict. Next, we used total serum IgG as the target and demonstrated that one could accurately model both the immune profile of an individual and their specific response to a disease. One might think that would be the hardest case, but actually polyclonal responses end up being only partially specific in aggregate, as they represent an average of a number of molecular interactions. We demonstrated there that, not only could we represent the overall immune profile, but we could accurately capture the information that distinguished the immune response of one disease from that of another and that the model was able to reject sequence independent noise, improving disease classification. Most recently we applied this to Lyme disease and showed that we could take the models of immune responses and project them onto the *Borrelia* proteome and again differentiate between cohorts but also identify both the known antigens used as current biomarkers as well as a number of new antigens. The current study takes the concept a step farther. Here we are using monoclonal antibodies with linear epitopes as the target because they are, in some sense, the ultimate in high sequence specificity binding systems. As in the past, we are using a sparse, essentially random sample of ~100,000 sequence-binding interactions to describe a much larger combinatorial space of about 16^{10} = a trillion possible sequences. Here we are challenging the system by using a set of targets with very high specificity and high affinity and asking whether the resulting model can recognize a known high affinity sequence. As stated above, the fact that it is a mAb is not really the point, though it does give the work a bit more potential practical relevance.

The reviewers have reasonably voiced some skepticism about whether we have really confirmed our hypothesis or not. Reviewer 3 formulated these concerns in the most specific and testable way. That reviewer pointed out that we were identifying 6mers and 7mers, not 10mers, and thus the space we are searching is smaller (17 million or 268 million, rather than something closer to a billion). Yes, the identified region is smaller than a 10mer, but the algorithm has to **find** the 6mers and 7mers in the 9

and 10mers because this is the only data it has to go on. Given an individual 10mer sequence and a binding value, the algorithm does not know which 6 amino acids were involved in the binding and thus which ones to value. It must learn that through comparison with many other sequences. In addition, there is really no evidence that the actual biological epitopes are confined to 6mers and 7mers. We know the region of the antigen it binds to contains those 6 or 7 amino acids, but that does not mean the context is unimportant. To deal specifically with the comments of Reviewer 3, we have selected larger regions of the antigen sequence from 6 to 10 residues and predicted the rank of those sequences out of a million random sequences of the same length. This is shown in Table S4 and S5 in the supplementary material. While there is some lowering of the rank in the longest sequences for 3 of the mAbs, it is pretty modest, and in 3 other cases, the rank actually improves in the longer sequences (the table is reproduced here for convenience). Note that in Table 2 of the main text now instead of showing 6mers and 7mers, we show all 7mers.

mAb	6 mers	7 mers	8 mers	9 mers	10 mers
Ab1	4.0 ± 0.6	3.0 ± 0.7	2.6 ± 0.4	15 ± 2	25 ± 4
Ab8	45 ± 10	7.9 ± 1.2	21 ± 7	68 ± 8	42 ± 16
4C1	22 ± 3	15 ± 1.5	5.4 ± 0.7	6.6 ± 0.7	3.4 ± 0.7
DM1A	6.6 ± 0.9	2.2 ± 0.5	1.4 ± 0.2	1.6 ± 0.2	1.8 ± 0.4
Lnkb2	1.0 ± 0.0	1.4 ± 0.2	5.2 ± 0.6	6.8 ± 1.2	17 ± 3
TF3B5	1.6 ± 0.4	1.1 ± 0.1	1.4 ± 0.2	1.8 ± 0.4	1.0 ± 0.0
C3	7.4 ± 1.8	9.5 ± 1.4	9.6 ± 1.3	2.2 ± 0.8	3.4 ± 0.6
9E10.3	42 ± 3	79 ± 4	13 ± 2	24 ± 3	33 ± 3

The point is that the model is able to find the binding motif in a 10mer with high specificity which means that it is not just memorizing a specific 6mer or 7mer (or 10mer), but actually recognizing what is important about the sequence of a 10mer. It has learned both what matters and what to ignore. That, in our view, validates the hypothesis, at least in the context of mAbs with linear epitopes. These results are described on p10 of the main Results and specifically addressed on p. 15-16 in the Conclusions.

What is it learning and what sequences is it learning from? There are also a number of reviewer comments that probe the issue of to what extent the algorithm is putting together sequence information from a large number of different sequences that are only partially related to the cognate sequence vs. relying on basically getting lucky and have a few sequences that result in high binding that largely dictate the ability of the model to recognize the cognate needle in a haystack. While related to the issue discussed above, it focuses more on similarity and number of sequences in the training set rather than on what size of sequence space the model is able to represent and is another important question. All of the reviewers suggested that we systematically remove the most similar sequences and retrain the models. Reviewer 3 made the related suggestion that we try training on smaller and smaller training sets (randomly reduced) and see where the models run out of information and fail, something we have done in two of our past publications (see above for links). We have done both here. Essentially what we are exploring is the fragility vs robustness of the models. We decreased sequence similarity to the cognate in steps, training on sequences sets with ≤ 6 amino acids in common with the 7mer epitope, then ≤ 5 , then ≤ 4 , ≤ 3 , and finally ≤ 2 amino acids in common with the 7mer sequence. The definition of “in common” used was having the right sequence and the right relative position, but occurring in any register within the 7mer (including registers where the cognate sequence and the comparator sequences only partially overlap). Thus, for SDLWKLL, the sequence RGRWADLGK would be considered to have 3 in common with the cognate (XXXXXDLXK) because the sequence order and spacing of the three matches, even though the register is shifted greatly. Table S5 in the supplementary

mAb	Number Sequences Removed to Achieve				
	≤6 matches	≤5 matches	≤4 matches	≤3 matches	≤2 matches
Ab1	0	0	3	150	3533
Ab8	0	0	7	257	4386
4C1	0	0	3	156	4005
DM1A	0	1	4	240	4688
Lnkb2	0	0	8	222	4381
TF3B5	0	1	10	334	6134
C3	0	1	26	543	7395
9E10.3	0	0	33	701	9075

material shows how many sequences were removed to achieve each reduced training set (shown below for convenience). Table S6 shows the ranks that resulted using the full model (weighting, shifting and compositional binding) trained on each reduced peptide set. Figure 4 in the main text presents this graphically (given below for convenience).

As you might expect, the more similar sequences removed, the worse the model is. However, even when one only allows at most three amino acids in common with the cognate, all but two of the mAbs remain in the top 0.1%. Thus at least in those cases, the model is clearly learning to piece aspects of the cognate sequence together. Ab8, on the other hand, is clearly quite fragile. To get to a point where there were no more than 4 matches, only 7 of the ~122,000 sequences were removed and this proved fatal to the model. Ab1 collapsed to worse than 1% when only 3 matches were allowed, although if you

Figure 4. Ranks of the 7mer cognate epitope (Table 1) in 1 million random sequences using training sets of array peptides in which sequences similar to the cognate sequence have been removed. The Y axis is the percent rank, with 0.0001% signifying rank 1 in a million. The X axis is the maximum number of amino acids that were in common with the epitope in the peptide array sequences used for training. The points and error bars represent the average and standard error of 5 randomly chosen sets of 1 million comparison peptides each and 12 randomly initiated models. In most cases, the error bar is too small to see in the plot.

allowed one to shift the cognate sequence by one position in the antigen sequence from RHSVVVP to FRHSVVV, it was back up in the top 0.1% again. This is presented on p. 13-14 of the Results and Discussed on p. 15-16 in the Conclusions.

Reducing the number of sequences in the training set tells a very similar story. This is now Figure S7 in the supplementary material. Again, as less sequence information (randomly chosen this time) is provided to train the algorithm, the lower the ranks of the mAbs in a million random sequences. And, again, the really fragile model is Ab8, which collapses with even a quarter of the sequences being removed. Note that each point is the average of 5 sets of 12 models and 5 randomly chosen peptide subsets to train on, thus even one

bad model due to key sequences being missing is going to drop the average rank a huge amount (the average of rank 1 and rank 500,000 is rank 250,000). In reality, sometimes Ab8 would give a much higher rank, just by chance. Again, the discussion of robustness is on p. 13-14 of the Results and p. 15-16 of the Conclusions.

Thus, we conclude that in most cases, the model has learned far more than just what a lucky sequence in the training set would have taught it and it has had to put the pieces together, learning what matters and what to ignore, rather than just memorizing a single pattern.

Notes on changes

In addressing the reviewer's comments, we realized that it would strengthen our study if we used the same length cognate

sequence (the same length of sequence taken from the cognate region of the antigen) for all of the mAbs. Thus, we decided to use 7mer sequences for all of them. In addition, reviewer 3 suggested, correctly, that it would be simpler for people to understand binding rather than the slope of the binding vs. concentration, so we went to using binding to calculate rank. This, however, meant redoing all the calculations. So, this changed Table 2 and Figure 2 (we had already used 7mer tiles for Figure 3). In the supplementary data, it changed Figures S4 and S6, removed Table S1 of the old manuscript (which gave binding values), and changed the sequences in Table S2. While the numbers are slightly different, the message has not changed.

Specific Responses (reviewer comments are in *italics*)

Reviewer #1 (Remarks to the Author):

In this paper a combined in-silico experimental pipeline is presented to map the linear epitope space relating to a particular antibody. I think it is a valuable study given that the authors can address the following comments:

1) *It is not clear to me whether antibody sequence information is taken into account. From my reading, each antibody model is trained separately on the linear epitope data. However there are also mentions of 'binding pairs' in the manuscript.*

The neural network is trained using a one-hot representation of the sequences of the peptides on the array as the input and the binding values due to binding of the mAb in question as the output. The linear epitope sequences are never included in the training set. These are what is being predicted by the models. The binding pairs consist of a sequence from the array and a set of binding values as a function of concentration of mAb. The trained neural network can then take any input sequence, whether used in the training set or not, and predict the binding to it. The binding measurements and mAbs are described briefly on p.5. The modeling is described on pages 6-9. Additional details are provided in the Methods and Supplementary Material.

2) I would emphasize the point that authors went 'the other way' to traditional methods that map abs with respect to an antigen. Mapping different antibodies with respect to an antibody has a clear discovery purpose as it gives novelty and priority to a typically known target. By contrast mapping epitopes to a known ab has a seemingly tenuous application. Chief potential application that is only scarcely mentioned is off-target binding as it is a big problem in contemporary immunotherapies. Authors could expand upon that.

As described in the general comments above, while it is true that this approach could potentially be used for analyzing off target binding (keeping in mind that for now we are only looking at linear epitopes) and finding potential targets of immune responses if you look more generally at serum IgG, as we have done with Lyme Disease (see above), our goal in this work is focused on exploring a more fundamental hypothesis about using sparse sampling of larger sequence spaces to create comprehensive models that recognized highly specific binding sequences (see above). That said, we do discuss potential applications, including off-target binding, on page 16 of the discussion.

3) Are there similar antibodies in the PDB to the ones mapped here? If so are their linear epitopes (or subsets of discontinuous) within the predicted sets?

It is not clear what the reviewer means by similar. As far as we are aware, these antibodies have not been co-crystallized with their targets. Note that these mAbs all bind to specific known linear epitopes.

4) There are many linear epitope prediction softwares (e.g. <http://tools.iedb.org/main/bcell/>) I think it would be worth it to either perform benchmarking or say why it does not apply here.

We reference a review on this subject now on p. 3 (ref. 18). However, there is a significant difference between what we are doing and the typical epitope prediction algorithms used. In particular, we start with binding data to a large number of random sequences and predict binding to any sequence, rather than either starting with a target molecule and asking what sequences are likely epitopes, performing docking calculations or tiling the sequences of the target. Again, we are largely using mAbs as examples of targets with highly specific binding sequences and using them to test the capabilities of the models we derived from sparsely sampled combinatorial space, though at least for mAbs with linear epitopes or significant linear binding regions, this could be used for prediction.

Reviewer #2 (Remarks to the Author):

In this work, Bisarad and colleagues investigate antibody binding against random peptide arrays. This

work is very interesting and rigorously carried out. I have a few comments.

5) have you considered data leakage in your model? That means, have you tried splitting the training/test data by sequence similarity? To quantify generalization of the predictor?

6) Are there peptides that are very similar in sequence yet have different binding? Are these predicted correctly? Can you make a plot where you plot delta peptide sequence similarity vs delta accuracy? or something like that?

(Answering 5 and 6 together) The issue of removing similar sequences was discussed in the general comments and resulted in making a plot of rank vs. similarity (Figure 4 of the new manuscript). As for whether there are similar sequences with different binding, that is what is shown in Figure 2 of the manuscript. Here predicted and measured data is shown for all single mutations of the cognate sequences of four of the mAbs. These sequences are clearly all very similar to the cognate sequence, but many result in huge binding changes. As you can see, while the predictions are not perfect, there is general correspondence between the predicted single amino acid changes from our models and the relative measured values. This is reflected in the correlation coefficients between them, which range from 0.60 to 0.87. Note that all of these values are normalized because the measured values for the single amino acid changes were done using a different set of arrays and different concentrations of mAbs than the arrays and concentrations used to build the model. However, in terms of relative values, they should be comparable. Figure 2 from the new manuscript is below for convenience:

7) have you tried baseline models which just take into account amino acid composition but not the sequence?

One of the approaches that we took in developing models for mAb binding was to separate the binding due to composition from that due to specific sequence. This was done by doing a simple linear fit to the composition using all the peptide sequences (one coefficient for each amino acid is multiplied by the number of that amino acid in the sequence). This was subtracted from the measured binding and the

remainder was considered the sequence specific part of the binding. The result of deconvolving the data into composition vs. sequence dependent binding is given in Figure S5 and described on p. 9. Figure S5 is reproduced below for convenience:

In most cases, the amount of binding that is strictly due to composition is small (it represents <0.1 contribution to the log of the binding in most cases). Figure S5 visually demonstrates any model taking into account only amino acid composition would be a poor predictor of antibody binding. 9E10.3 shows one of the larger amounts of variation due to the compositional component and here dealing with this component and the sequence dependent binding separately in the model resulted in a more highly ranked prediction for the cognate sequence binding.

8) Are there differences prediction performance between the antibodies? if so, why?

As shown in Table 2 and the new Table S5 in the revised supplementary material, the model ranked all of the mAbs as very high cognate epitope binding compared to 1 million random sequences, but there are clearly differences. This includes marked differences in how robust the different models are, as alluded to in the general comments above, and in Figures 4 and S7. The “why?” is somewhat speculative, but there are key differences in the amount of data that binds significantly to the different mAbs as shown in Figure 1, where you can see that some mAbs, such as Ab1 and Ab8, simply have very few high binding peptides on the array, and thus few high information sequences. This is presumably what results in the more fragile models, as you can see in Figures 4 and S7 (reproduced in the general comments for convenience). TF3B5, on the other hand, has rather strong binding sequences on the array, and a very robust prediction of high binding by the cognate sequence, again as can be seen in Figures 4 and S7. The substitution matrix of Figure 2 (also shown above) gives additional clues; some of the mAbs require very exact amino acid sequences and others are more tolerant of substitutions. All of this is discussed on pages 13 and 14.

9) Can you add an overview Fig 1 of the experimental/computational setup?

The experimental approach is simply a matter of binding a small volume of sample to an array of peptides on a silica surface. There is not much to diagram there. The neural network is very simple and has been diagrammed in one of our previous papers. Please see Figure 1 of Ref. 6.

10) Did you perform controls (randomize peptide sequence  does prediction accuracy go to random?)

This is now given as Table S3 in the supplementary material. This is referred to on p. 9 of the main text. The results are nearly random as expected.

mAb	Rank
Ab1	420,000 ± 170,000
Ab8	140,000 ± 40,000
4C1	310,000 ± 130,000
DM1A	300,000 ± 130,000
Lnkb2	370,000 ± 150,000
TF3B5	420,000 ± 130,000
C3	610,000 ± 100,000
9E10.3	530,000 ± 90,000

11) Please provide numbers/percentages in figure 1 as to how many peptide are above noise.

To do this we had to quantitatively define “noise” or perhaps more accurately background level, since peptides that don’t bind are not noise in that they contribute to building the model. Then we had to quantitatively define how far above a value needs to be in order to be counted. What we did was determine how many peptides at each concentration and each mAb were more than 2 standard deviations in that peptide’s value (determined from the four replicates taken at each concentration and each mAb) above the median value for the whole dataset for that concentration and mAb. Figure 1 is very busy, so we put this data in Table S1 in the supplementary material and referenced it on p. 5 of the main text. It is reproduced here for convenience (also referenced in the legend to Figure 1):

Table S1: Number of Peptide Sequences >2 Standard Deviations above the median binding value

mAb	Number of Sequences Above the Median			
	0.125 nM	0.5 nM	2 nM	8 nM
Ab1	3	23	89	195
Ab8	0	1	2	9
4C1	89	123	364	1146
DM1A	19	69	321	101
Lnkb2	30	65	116	69
TF3B5	208	584	8428	7323
C3	29	48	178	228
9E10.3	28	111	5	13393

The increase in numbers is not always very consistent at high concentration. This is because there is a balance between how large the standard deviation was on the array vs. the level of the median. For some reason, 9E10.3 had an array at 2 nM that was overall at lower scale. This was averaged in because the correlation to the other arrays was fine, it was just scaled lower, but we chose not to try and manipulate the data scale since we wanted to preserve as much as possible the concentration dependence. That array resulted in a big standard deviation for

the peptides at that concentration and thus few peptides that were counted as >2 standard deviations above the background value.

Reviewer #3 (Remarks to the Author):

Summary

Bisarad, Kelbauskas, Singh et al. present a dataset of eight antibodies with binding data against 122k randomly designed peptides. They show that, for each antibody, a simple Multilayer Perceptron (MLP) trained on this data consistently ranks the original (cognate) linear epitope highly among 1m different random peptides of the same length. The authors hypothesise this high throughput screening and ML model could be useful for selecting highly specific mAbs for therapeutics or diagnostics.

12) The paper is clearly written but lacks a strong argument linking the method results (high ranking of the cognate epitope) with the original hypothesis, and the use case and impact of the work is not clear. The paper also lacks robust analyses in places. We have highlighted some of our main concerns below.

Please see the general comments for a clearer description of the manuscript’s hypothesis and goal which is now hopefully better laid out in the introduction and address at the beginning of the conclusion.

Major

13) The paper does not address the question of whether or not the high throughput screening and ML model can identify polyspecific binding. Instead, the paper focuses on the ability of the trained MLP to rank the cognate epitope highly among a random background. This result is not clearly linked to the original question. Though many random peptides were also ranked highly by their model (above the cognate epitope), the authors did not test their binding experimentally.

Our hypothesis is now hopefully more clearly stated on p. 2-3, and discussed in the general comments.

With regard to providing experimental evidence that we can predict polyspecific binding, as shown in Figure S4, also given here for convenience), the models could generally predict the measured binding to the vast majority of the peptides on the array, at least in those mAbs for which there were substantial strong binding peptides. The measured data is experimental data and the binding is polyspecific. The predictions are real predictions because the peptides predicted in the 10-fold cross validation were not included in the training. In those mAbs with substantial strong binding on the array, prediction was quite good. Similarly in Figure 2 (provided above in response to reviewer 2) all possible single amino acid changes were made to each

cognate sequence and were both predicted and measured. Again, this is experimental data compared to binding predictions for sequences not used in the training and again shows that the model does a reasonable job of predicting which sequences will bind strongly and which will not.

These ideas and the relationship between our data and our hypothesis is further discussed in the general comments above and further elaborated on in the conclusions of the paper.

14) In Figure 3 (page 11) the authors consider the entire protein antigen of each mAb, break it down into peptide chunks, and rank each chunk using their trained MLP. Encouragingly, the cognate epitope ranks highly. However, if the antigen were known a priori (as is often the case) and the goal was simply to identify any linear epitope, then would the more efficient experiment not be to select the e.g. 200 or 1k peptides that comprise the antigen and test for binding against these instead of 122k random epitopes?

Two points to consider. First, as described above, identifying epitopes is a possible use of this technology, but it is not really the fundamental point of the manuscript. That said, yes tiling of a known antigen can be done and has been done many times before, but it requires a custom set of peptides be made for every antigen. The benefit of the random array approach described here is that it provides a single, common, commercially available, array that you could bind any antibody to. More importantly it provides a model you can apply to any sequence. For example, if you are wondering if your SARS-CoV-2 spike protein mAb will bind to a new variant of the virus, you don't need to make a new array or even do another experiment. The model you have predicts for any sequence. We hasten to point out, however, that the arrays and models used here do have limitations in this regard. This particular array uses only 16 amino acids, and the models do not yet incorporate information from protein structure that will be needed to move beyond linear epitopes. These are the things our lab is working on now from a practical perspective, but the current study says that the fundamental hypothesis is valid... we can derive very specific information about sequence binding from arrays of random sequences binding to targets with very specific binding sequences. That is the foundation that makes it possible to move to the next level of complexity. The practical and fundamental value of this work is discussed on pages 16-17.

15) Assuming the goal of the paper is actually focused on ranking the cognate epitope highly (this should be justified), then more robust statistics should be provided here. Given final comparisons are only made to peptides of the same length as the cognate epitope (6-7 amino acids) and only 16 amino acid types are allowed, the total search space is 16^6 or 16^7 (~17m-268m). If the cognate epitope were not known a priori, the chances of selecting the cognate epitope in each instance given 1m designs is 6% or 0.4%. To be robust, the authors should evaluate how highly the cognate epitope ranks amongst the entire possible sequence space - this should certainly be computationally possible for length-six designs at least. If however, the emphasis remains on predicting poly-specificity, with ranking of the cognate epitope being a measure of evidence for this, then this should be more clearly emphasised and argued for.

This is an important issue and is largely dealt with in the general comments above and discussed in the text on p. 15-16. Note that Table 2 is now showing only 7mer cognate sequences (not 6 or 7), demonstrating one can stick with one tile size for all the mAbs. To take that further, 6-10mer sequences from the antigen sequences were used in Tables S3 and S4 showing that the tile size has only a modest

effect on the ability to recognize the cognate region (sometimes improving rank a little and sometimes decreasing rank a little). The reviewer is correct that these are important issues to explore and the addition of these new figures strengthens the overall argument of the paper.

There is poly-specificity of the models, something that you can see in Figure 2 with the point mutations and Figure S4 with the diverse peptide sequences of the array (at least for the mAbs which have reasonable binding over a range), previous publications from our group have already shown that for other samples. That said, what we really wanted to test with these mAbs was whether the sparse, unbiased sampling could be used to train a model that would be able to recognize a known interaction of a target with a particular amino acid sequence that is extremely specific, which is why we are using mAbs that have linear epitopes. Measuring weaker protein peptide interactions or interactions of many IgG molecules with a target represents an average of multiple interactions. Here we are exploring recognition of a particular interaction.

As for statistics, in Table 2 the whole process is performed over 120 times for each of the 8 mAbs, with 5 sets of 12 independently derived models for an mAb each applied to 2 unique sets of 1M random sequences. Thus, the rank is statistically very well known. Clearly it is not going to change appreciably between sampling 10 million and sampling 268 million. It is important to both use multiple models (which have random starting points) and multiple sets of random sequences, as we have done, to make sure we have considered both sources of variability of the rank. While one could do all of the 268 million possibilities for one model and one mAb, doing it for many models and 8 mAb is a lot of computation that does not gain anything statistically.

16) However, to be yet more robust, it should be assumed that the cognate epitope length is also not known a priori (peptides of length 5-11 are used for training). In this instance, the cognate epitope should be evaluated against designs of random lengths. Here, the sequence space expands to ~1 trillion (infeasible to exhaustively search), as stated in the paper (page 2), and the probability of selecting the cognate epitope by chance given 1m random designs is just 0.0001%. Again, it should be considered whether or not selecting the cognate epitope in the top X designs is the true goal of the paper.

Again, see Tables S3 and S4 and the general comments above. Hopefully it is now clear what we were trying to achieve here and why we chose to focus in this manuscript on mAbs and the ability to recognize their specific sequences.

17) Only one MLP model is considered for training. It would be useful to try alternative architectures such as a random forest or convolutional neural network (CNNs) for comparison. CNNs would be particularly suited to the task due to their ability to pick up position-independent motifs. In addition, the authors do not describe if/how they optimised their MLP architecture or training hyperparameters. The number of training epochs or iterations is also not stated.

To address the last question first, one advantage of a relatively simple neural network model such as the one used here is that it is fairly easy to optimize with relatively few hyperparameters to tune. We have performed optimization of the numbers of hidden layers and nodes as well as experimenting with different learning rates, number of epochs trained, minibatch size, dropout percentages, etc. This is now noted on p. 18 of the methods. As for CNNs, we have used CNN models. They work, but in terms of the kind of prediction discussed here (continuous, linear epitopes), have thus far not been shown to

give predictions any more accurate than the simpler models. Thus, we used the simpler fully connected model, which appears to be robust (you can vary number of layers and nodes considerably with similar results). Where CNNs, and perhaps even transformer models, may be more useful is as we start to try and predict the binding of discontinuous epitopes that, by definition, depend on the interaction of many small pieces over larger regions of sequence.

18) The significance of the steps taken to improve the MLP performance - oversampling positive sequences given the large class imbalance, and "shifting" the sequences - are overstated. The paper says "The biggest advance in the model came from forcing the neural network to ignore the absolute position of the sequence in the vector and generalize its learning so that it recognized the sequence regardless of its position within the artificial register of the input vector. This form of data augmentation by training on translational variants is well-established to help generalize convolutional neural networks for image recognition tasks. It is remarkable in many respects that the much simpler neural network model used here (2 hidden layers of 250 nodes, fully connected) would be able to utilize this data augmentation to generalize in this way." These training techniques are standard, necessary, and not as remarkable as claimed.

Our intention was simply to point out what mattered in the improvement of the apparent accuracy of the predictions for these models and this dataset (weighting, shifting of sequence register, removal of compositional binding), not to claim these as fundamentally new algorithms. That said, what we are doing, as the reviewer points out above, is not a CNN and really rather different. CNNs look at pieces of the sequence. This always considers the whole sequence, just in different registers. However, we have used language on p. 15 to make it clear that we are not claiming a new algorithmic breakthrough, but only describing what was required to enhance the predictive capability.

19) Given the large training set size (122k), lack of sequence identity filtering, proven sequence similarity of the top hits (Figure 1), and oversampling of these top hits, the MLP appears to be simply memorizing its input and struggling to predict anything out of distribution. It would be useful to repeat the training with strict sequence similarity cut-offs in the train and validation datasets - this would help justify the claimed benefits of the sparse sampling strategy. Additionally, models with decreasing training set sizes could be trained to determine the minimum training set size to still result in a high ranking of the cognate epitope.

This was discussed as part of the general comments and in Figures 4 (rank in models trained with similar peptides removed) and S7 (rank in models trained with decreasing total numbers of array peptides). At some level, memorization and learning by example are closely related, but what one can conclude is that the model has both learned, in a piecemeal fashion, what amino acids are important at which relative positions and which are not as important. This issue is now discussed on p. 15-16.

20) Why were the cognate epitopes of TF3B5, C3, and 9E10.3 not included in the 122k peptides tested for binding? There must have been space on the array and their exclusion is not explained, simply stated.

In order to make these arrays cheaply and in large numbers, we need to keep the total number of synthetic steps low. In principle, to be able to make ANY arbitrary set of 11-mers on the array by stepwise synthesis would require the use of all 16 amino acids at each residue, so a total of $16 \times 11 = 176$

total photolithographic steps. While not impossible, this greatly reduces the ability to make large numbers of very high-quality arrays. So instead, we designed near-random arrays using 64 steps, but this limits the number of specific sequences you can design into the array. Note that TF3B5 and 9E10.3 were synthesized on a separate array with their single amino acid variants (Figure 2), but this was a different array with completely different sequences, and we did not want to do a direct comparison of binding values between this and the arrays used for the rest of the work. Thus, we did not include the values of the cognate binding from these other arrays in Figure S4. Figure 2 is based on relative values. This is now stated in the legend to Figure 2.

21) Correlation coefficients and the gradients of the lines of best fit should be added to Figure S4 to better understand how well the predicted and measured log10 values agree.

Done and we repeated the predictions multiple times to give better statistics, See Figure S4.

22) The language used to describe neural network training and architecture could be improved for clarity and to remove unnecessary complexity, e.g.:

- *“The matrix manipulations in a neural network are register-specific; an amino acid in the first position always “sees” the same set of processes take place as the information moves through the neural network.”*
- *“it recognized the sequences regardless of its position within the artificial register of the input vector.”*

Modified, see p. 8-9

Minor

23) The Introduction states “Binding typically involves interactions between the mAb and 5-8 amino acids and in a specific spatial arrangement”. However, for full-length proteins, which this paper considers, the full epitope is often composed of multiple linear/conformational epitopes, typically totalling 10-20 residues (Reis et al., 2022). This fact should be clarified here.

We are referring to linear epitopes here rather than discontinuous ones. We have generalized the statement so that it no longer quotes a particular number. See p. 3

24) Figure 3's image quality is very low and should be improved.

Corrected

25) Ideally, should additional epitope regions be identified in Figure 3 that are close in space to the cognate linear epitope (if they exist)?

The coloring is automatic/objective and based on binding and is continuous between green and red. Indeed it is on a log scale, making even weak binding quite evident. If you look carefully at Ab8, for example, you can find a couple of regions that are “off-green” and represent a little binding. 9E10.3, as pointed out in the paper, is the poster child for having multiple very similar sites and there you see binding to 3 regions. We have clarified this on p. 13.

26) Given the greater interpretability and similar performance, why was the concentration maximum (Table S1) not used in place of the slope value (Table 2)?

Done, we no longer are using slope for any of the calculations.

27) Can the similarity between the predicted and measured values in Figure 2 be quantified in some way?

Correlation coefficients between the measured and predicted values are now added to Figure 2.

28) In the methods section “since neither the cognate sequences or any of the million random sequences ... were part of the training” should be amended to ‘since neither the cognate ... were in the training set’.

Modified

REVIEWERS' COMMENTS:

Reviewer #1 (Remarks to the Author):

The authors responded to my comments.

Reviewer #2 (Remarks to the Author):

The authors have addressed all of my comments.